# Ultra-depleted hydrogen isotopes in hydrated glass record Late Cretaceous glaciation in Antarctica

Demian A. Nelson[1], John M. Cottle [1 ✉], Ilya N. Bindeman [2] & Alfredo Camacho[3]

The Early Jurassic Butcher Ridge Igneous Complex (BRIC) in the Transantarctic Mountains contains abundant and variably hydrated silicic glass which has the potential to preserve a rich paleoclimate record. Here we present Fourier Transform Infrared Spectroscopic data that indicates BRIC glasses contain up to ~8 wt.% molecular water ($H_2O_m$), and low (<0.8 wt.%) hydroxyl (OH) component, interpreted as evidence for secondary hydration by meteoric water. BRIC glasses contain the most depleted hydrogen isotopes yet measured in terrestrial rocks, down to $\delta D = -325$ ‰. In situ $^{40}Ar/^{39}Ar$ geochronology of hydrated glasses with ultra-depleted $\delta D$ values yield ages from 105 Ma to 72 Ma with a peak at c. 91.4 Ma. Combined, these data suggest hydration of BRIC glasses by polar glacial ice and melt water during the Late Cretaceous, contradicting paleoclimate reconstructions of this period that suggest Antarctica was ice-free and part of a global hot greenhouse.

Hydrogen isotope studies of altered rocks and minerals have enabled identification of cold climate and glacial conditions[1–3]. These studies demonstrate that ancient glacial conditions can be recorded and preserved in δD values within volcanic glass and secondary minerals, and encourage exploration for previously undocumented glaciation in the geologic record.600-m-high.

Antarctica has been in a polar position, near the South Pole, since the late Paleozoic[4] and provides an ideal location to explore the potential for hydration of volcanic glass by isotopically depleted polar waters. In particular, the Butcher Ridge Igneous Complex (BRIC) of the Early Jurassic Ferrar Large Igneous Province (LIP) in Antarctica (Figs. 1, 2) contains the southernmost occurrence of abundant volcanic glass that may have been exposed to polar glacial melt water since emplacement c. 183 Ma[5].

The BRIC is best exposed along a 10-km-long, escarpment at the head of the Darwin Glacier, ~280 km southwest of McMurdo Station, Antarctica[5] (Figs. 1, 2). The BRIC preserves abundant felsic volcanic glass within a 1-km-long dacitic vitrophyre pod unit and a well-layered rhyolite unit, both of which contain distinct meter-scale layering (Fig. 2). These unique layering features are interpreted to result from

syn-emplacement alteration, hydration, and devitrification processes[5]. Detailed petrographic observations are presented in ref. 5, and briefly summarized here. The vitrophyre pod unit contains plagioclase, pyroxene, and oxides set in a glass matrix with veins of alteration. The well-layered rhyolite unit consists of vitrophyre, crystalline, and marginal layers (Fig. 2). Vitrophyre layers have similar mineralogy to the vitrophyre pod but include potassium feldspar. Crystalline layers are holocrystalline resulting from alteration and complete devitrification of glass within vitrophyre layers to secondary potassium feldspar and quartz with abundant oxidation observed in thin section[5]. The marginal layers represent an intermediate stage of devitrification between vitrophyre and crystalline layers. Marginal Layers are holocrystalline in hand sample but preserve a minor amount of heavily altered glass matrix when observed by scanning electron microscope and contain abundant alteration and devitrification textures[5]. The remaining units of the BRIC are massive or diffusely-layered and are almost entirely holocrystalline containing primarily plagioclase, pyroxenes, potassium feldspar, and oxides.

Here we present the most depleted hydrogen isotope compositions measured in terrestrial rocks and minerals so far, down to −325 ‰

[1]Department of Earth Science, University of California, Santa Barbara, CA, USA. [2]Department of Earth Sciences, University of Oregon, Eugene, OR, USA. [3]Department of Earth Sciences, University of Manitoba, Winnipeg, MB, Canada. ✉e-mail: cottle@geol.ucsb.edu

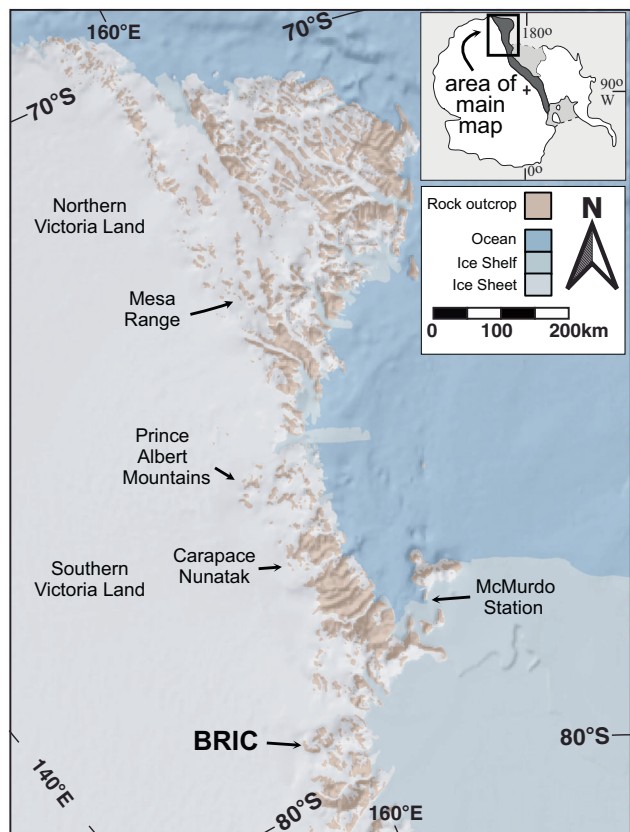

**Fig. 1 | Location of Butcher Ridge Igneous Complex.** Bedrock map of Northern and Southern Victoria Land, Antarctica, showing the distribution of outcrop and location of the Butcher Ridge Igneous Complex (BRIC) and other localities referred to in the text with demonstrated mid to late-Cretaceous alteration ages. Basemap constructed using Quantarctica v3.2 from the Norwegian Polar Institute (Matsuoka, K. et al. Quantarctica, an integrated mapping environment for Antarctica, the Southern Ocean, and sub-Antarctic islands. Environmental Modeling and Software 140, 105015 (2021)) with Rock outcrop data from SCAR Antarctic Digital Database (ADD) Version 7.0.

δD in BRIC glasses that contain abundant (2–6 wt.%) molecular water. We also present bulk-rock water content and in situ water distribution ($H_2O$ and OH) mapping using Fourier Transform Infrared Spectroscopy (FTIR). These data and in situ $^{40}Ar/^{39}Ar$ geochronology suggest that the extremely depleted δD values reflect incorporation of polar glacial melt waters during a Cretaceous hydration alteration event at 72 to 105 Ma and a peak at c. 91.4 Ma. The presence of polar glaciation during this time, i.e., Cenomanian through Campanian, contradicts existing paleoclimate models indicating Antarctica was ice free due to a thermal maximum and global hot greenhouse conditions[6–12]. Instead, these data support the presence of widespread polar glaciation during the Late Cretaceous[13–19].

## Results

### In situ water content and speciation

The total water content of volcanic glass ($H_2O_t$) consists of water incorporated as hydroxyls (OH) and molecular water ($H_2O_m$). A complex interplay between interface kinetics, diffusion, and re-speciation processes controls the speciation of water involved in secondary hydration of degassed silicate volcanic glass[20–23]. Generally, water incorporated as OH is interpreted as magmatic in origin and retained from earlier incomplete degassing, while molecular water is interpreted as the primary diffusing species and sourced from paleo-environmental meteoric waters[24]. However, there is evidence that magmatic water may be incorporated as $H_2O_m$ and meteoric water

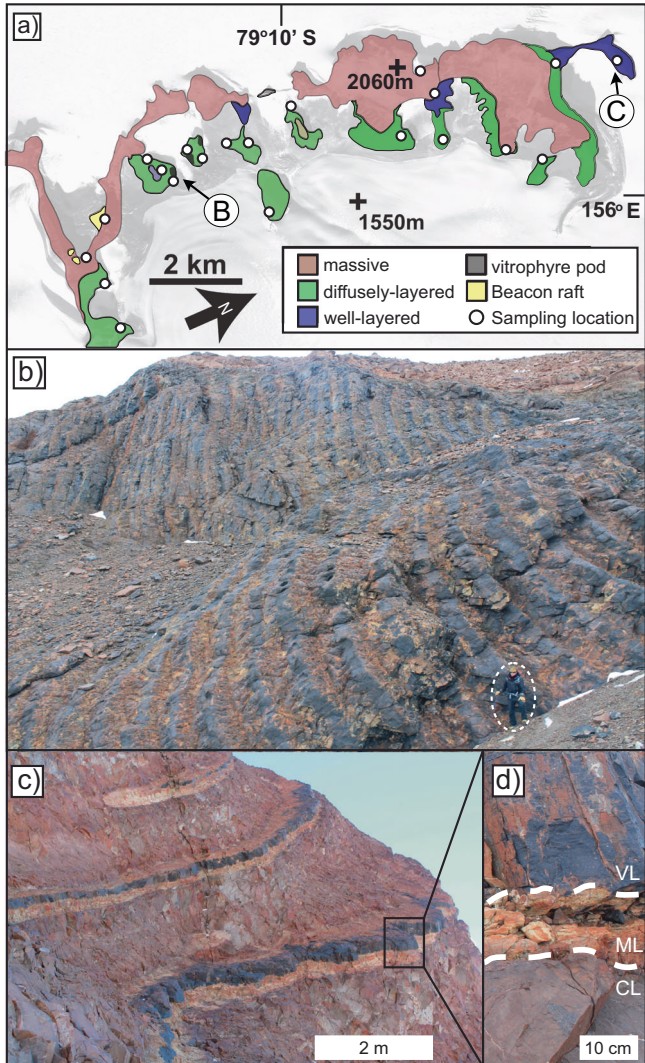

**Fig. 2 | Geological features of the Butcher Ridge Igneous Complex. a** Simplified geologic map of the Butcher Ridge Igneous Complex (BRIC) with locations of images (**b**) and (**c**) arrowed (overlain on WORLDVIEW-2 satellite map) Modified from Nelson et al. [5]. **b** Vitrophyre pod unit of the BRIC, person for scale lower right circled (**c**) well-layered rhyolite unit, and (**d**) close-up showing the distinct vitrophyre (VL), marginal (ML), and crystalline (CL) layers. Images modified from: Demian A. Nelson, John M. Cottle, Blair Schoene; Butcher Ridge igneous complex: A glassy layered silicic magma distribution center in the Ferrar large igneous province, Antarctica. GSA Bulletin 132(5–6), 1201–1216 (2020). https://doi.org/10.1130/B35340.1.

incorporated as OH depending on the degassing and secondary hydration history of the glass[25]. In either case, the hydrogen isotope composition of hydrated glass represents a mixture of at least these two isotopically distinct sources that may be distinguished based on water speciation. We measured the in situ water content and speciation (i.e., molecular vs. hydroxyl) of BRIC glass using Fourier Transform Infrared (FTIR) spectroscopy on doubly polished chips (see analytical methods below) to determine if the BRIC contains molecular water from a paleo-environmental meteoric source and to assist in deconvolving the isotopic source(s).

FTIR water content maps (Fig. 3) indicate that glass within the vitrophyre pod has high molecular water contents, ranging between 3 and 8 wt.% $H_2O_m$, relatively low OH contents up to 0.8 wt.% OH and higher molecular water content correlating with lower hydroxyl content. Vitrophyre layers from the well-layered rhyolite unit also have high molecular water contents ranging from 2 to 4 wt.% $H_2O_m$ and low

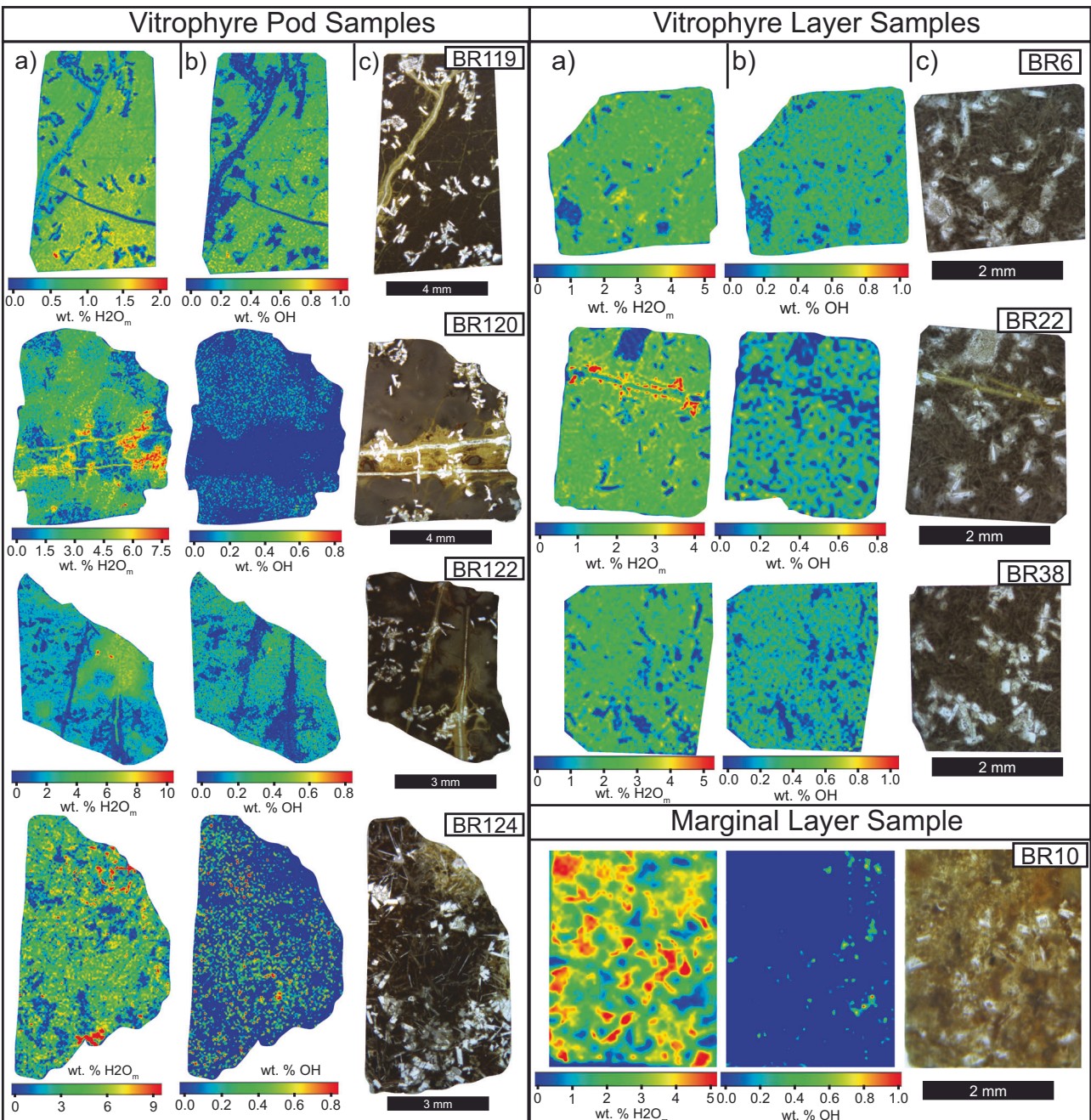

**Fig. 3 | Fourier transform infrared spectroscopic data for Butcher Ridge Igneous Complex rocks.** Fourier Transform Infrared Spectroscopy (FTIR) water speciation map data for vitrophyre pod, vitrophyre layer, and marginal layer samples of the Butcher Ridge Igneous Complex. **a** Molecular water (H2Om) content, (**b**) hydroxyl (OH) content, and (**c**) transmitted light image.

OH contents of up to 0.8 wt.% OH. A sample of the marginal layer of the well-layered rhyolite unit has patches of high molecular water, 2 to 5 wt.% $H_2O_m$, with no detectable OH. These water contents are consistent with solubility experiments for $H_2O$ in rhyolitic glass of 3–5 wt.% at 175–375 °C[26] and indicate comparable or lower temperatures were likely involved during secondary hydration and alteration of the BRIC.

These textural observations, water content, and speciation data demonstrate that BRIC glasses preserve high amounts of molecular water and low hydroxyl content, consistent with secondary hydration and minimal preservation of magmatic water or incorporation of secondary waters as OH. Therefore, hydrogen isotopes incorporated into BRIC glass during secondary hydration likely contain critical paleoclimate information.

## Bulk water and hydrogen isotopes
We measured the bulk-rock water contents and hydrogen isotope composition (analytical methods described below, Supplementary Data 1, Fig. 4) of samples of the vitrophyre pod, vitrophyre layers, and marginal layers to constrain the δD value of the incorporated paleowater as a paleoclimate indicator (Methods as described in Martin et al. 2017[27]). We also measured the bulk water and δD values of crystalline layers of the well-layered, diffusely-layered, and massive units to determine the magmatic δD values to deconvolve mixing between environmental and magmatic waters. Holocrystalline BRIC samples have an average δD value of −115 ‰ ($n = 9$) and vary from nearly primary magmatic (−88‰) to −140 ‰ δD with increasing bulk water content (average of 0.78 wt.% $H_2O_t$), characteristic of

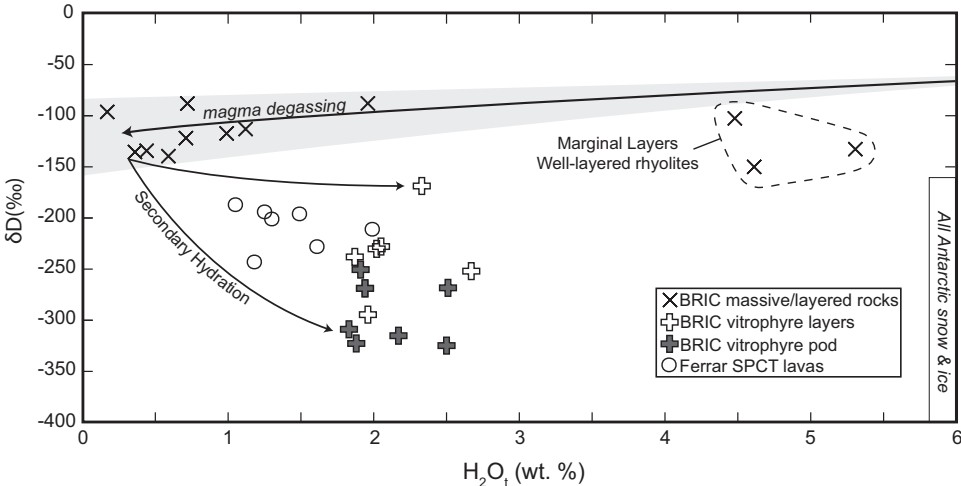

**Fig. 4 | Hydrogen isotope and bulk water data for Butcher Ridge Igneous Complex rocks.** Hydrogen isotope and bulk water data for samples of the Butcher Ridge Igneous Complex (BRIC), Scarab Peak Chemical Type (SPCT) lavas, and Antarctic snow and ice[28,29].

fractionation during degassing[6]. Despite the mineralogy of crystalline samples being dominated by plagioclase, pyroxene, oxides, and secondary devitrification minerals potassium feldspar and quartz, water within crystalline samples is interpreted to be stored within hydrated secondary minerals or trace hydrated glass[5]. Vitrophyre pod samples, in contrast, have a much higher bulk water content (2.13 wt.% $H_2O_t$ average) and lower average δD values ranging from −250 to −325 ‰ with an average of −298 ‰. Vitrophyre layers from the well-layered unit have similar bulk water contents as vitrophyre pod samples (2.12 wt.% $H_2O_t$ average) but intermediate δD values ranging from −169 to −295 ‰ and an average of −240 ‰. Marginal layers have the highest bulk water contents averaging 4.8 wt.% $H_2O_t$ but high δD values ranging from −104 to −151 ‰ and an average of −130 ‰.

To establish the source of the water, $\Delta'^{17}O$ values were measured in the least-hydrated ($\Delta^{18}O$ = 6.78, $\Delta'^{17}O$ = −0.065 ± 0.011 ‰) and lowest δD samples ($\Delta^{18}O$ = 9.24 ‰, $\Delta'^{17}O$ = −0.087 ‰) following the methods of Bindeman (2021)[30] (Supplementary Data 2). Samples plot on the terrestrial mass fractionation line, and the end-member δD values in BRIC glasses are interpreted to reflect the δD of incorporated environmental water at the time of secondary hydration[31].

BRIC glasses have considerably more depleted δD values than any other known terrestrial rocks and reflect δD values of incorporated environmental water[31]. Importantly, there is no evidence that significant kinetic hydrogen isotope fractionation (identifiable to within ±20 ‰ δD) occurs when molecular water is mobilized through volcanic glass[32]. Fractionation does occur during uptake of environmental water, such that δD values of the secondarily hydrated volcanic glass may be offset by −33 ‰ from the environmental water[24,25], however, even lower $\Delta D_{glass-water}$ values down to −90‰ in some higher temperature hydration experiments have been observed due to secondary repartitioning of environmental water into the isotopically depleted OH site[7]. Consequently, the δD values recorded in the BRIC glasses record paleo-environmental waters isotopically depleted down to approximately −292 ‰ δD, or even if all water has repartitioned into OH, which is not evident in the FTIR data (Fig. 2), the original paleo-environmental water would still be very depleted down to −235‰. It is likely, however, that the bulk-rock measurements include a minor contribution of higher δD magmatic water. Therefore, the original paleo-environmental δD values are expected to have been more depleted than calculated here. Furthermore, secondary clays and micas have not been observed in BRIC vitrophyre samples but even if present, they would not

significantly change the inferred δD of environmental water because mica-water fractionations are comparable to that of glass-water, i.e., ΔD ≤ −100‰[33].

Such low δD values from −235 to −292 have only been observed in polar snow and ice[28,29]. Recent glaciation in Antarctica began ~34 Ma and persists to the present with snow and ice ranging from −150 to −450 ‰ δD[28,29,34] (Fig. 3). The most recent glaciation in Antarctica is therefore a potential source for hydration of the BRIC, but older glacial episodes since the Jurassic may also have been responsible. We addressed the timing of hydration using $^{40}Ar/^{39}Ar$ geochronology.

## In situ $^{40}Ar/^{39}Ar$ geochronology

Secondary hydration of glass by ~2–5 wt.% water or greater has been shown to disturb the $^{40}Ar/^{39}Ar$ system by the substitution of $H^+$ for alkalies, allowing potassium migration out of the glass structure, and/ or alteration of the glass structure to promote loss of argon[35,36]. Argon isotope systematics in altered glass may be further complicated by uptake of atmospheric argon, excess non-radiogenic, and/or non-atmospheric argon as well as possible kinetic mass fractionation of argon during hydration[35]. We anticipate hydration occurred below the closure temperature of argon within glass and minerals, i.e., <500 °C[26], and, therefore, hydration is the principal mechanism for disturbance of the $^{40}Ar/^{39}Ar$ system. If hydration of the BRIC glass favored loss of potassium then the $^{40}Ar/^{39}Ar$ age would be older than the emplacement age (~183 Ma). If hydration favored argon loss, then the $^{40}Ar/^{39}Ar$ age should be younger than the emplacement age and consequently reflect a partial or complete resetting of the $^{40}Ar/^{39}Ar$ age to the age of hydration involving polar glacial met water[37,38].

We utilized in situ $^{40}Ar/^{39}Ar$ geochronology (see methods section below) targeting hydrated glass within six BRIC samples to determine if samples experienced resetting of the argon system during secondary hydration by meteoric waters (Fig. 5, Supplementary Data 3, 4). Analyses were performed on four vitrophyre pod samples, one vitrophyre layer and one marginal layer from the well-layered rhyolite. Vitrophyre pod samples yield similar $^{40}Ar/^{39}Ar$ dates with an age range of 169.8 to 72.1 Ma and a peak age of c. 91.4 Ma ($n$ = 41). The sample of the vitrophyre layer from the well-layered rhyolite unit yielded a date range of 190.0 to 137.1 Ma ($n$ = 5). Glass from vitrophyre samples, layers and pods are spatially heterogenous and variable within each sample and, therefore, a precise single age cannot be determined. We suspect some analyses inadvertently included a variable contribution of phenocrysts, with original $^{40}Ar/^{39}Ar$ age c. 183 Ma, and hydrated glass with much younger dates and accounts for the data for vitrophyre dates

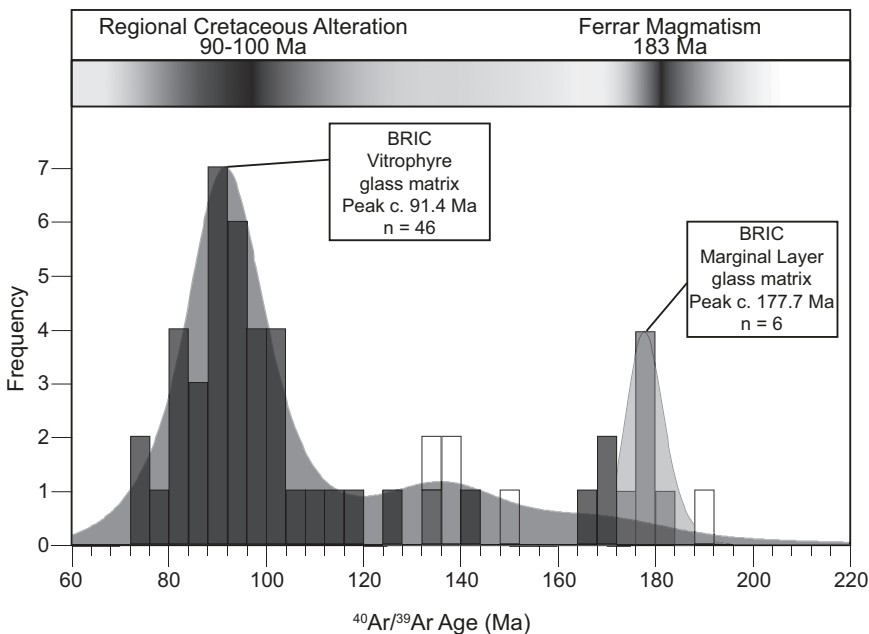

**Fig. 5 | Ar/Ar geochronology data for Butcher Ridge Igneous Complex rocks.** Kernel density plot and histogram of in situ $^{40}Ar/^{39}Ar$ geochronology data for vitrophyre pod samples (dark gray), the vitrophyre layer sample (white), and the marginal layer sample (light gray), BR12-10. Fields for previous regional Cretaceous alteration ages in the Transantarctic Mountains (TAM) from 90 to 100 Ma[39,40] and the Ferrar Large Igneous Province, 183 Ma, are provided for comparison.

that fall between ~105 and 190 Ma. Consequently, dates for vitrophyre glass from ~72 to 105 Ma with a peak at c. 91.4 Ma provide the most reliable alteration ages determined for vitrophyre pod samples. The marginal layer of the well-layered rhyolite yielded dates from 182.8 to 175.7 Ma and a peak age of c. 177.7 Ma ($n$ = 6), close to the emplacement age and suggest minimal resetting. In summary, $^{40}Ar/^{39}Ar$ geochronology of altered and hydrated marginal layers indicate an Early Jurassic age consistent with emplacement during Ferrar LIP magmatism and syn-emplacement alteration, hydration, and layer formation[5]. The Late-Cretaceous, ~72–105 Ma with a peak at c. 91.4 Ma, dates for the vitrophyre pod samples indicate hydration favored argon loss and partial to complete resetting of the original Jurassic $^{40}Ar/^{39}Ar$ age and provides our best estimate for the age of secondary hydration by polar glacial melt water.

## Discussion

Two observations are emphasized from the combined FTIR, hydrogen isotope, and in situ $^{40}Ar/^{39}Ar$ geochronology data: (1) The BRIC experienced hydration/alteration during the Cretaceous, ~72–105 Ma with a peak at c. 91.4 Ma; and (2) Cretaceous hydration of the BRIC involved polar glacial melt water. Importantly, the age of Cretaceous alteration recorded in vitrophyre pod samples is indistinguishable from regional Cretaceous alteration previously documented by $^{40}Ar/^{39}Ar$ geochronology of secondary apophyllites (Fig. 6) in other Ferrar LIP rocks at various locations along the TAM[39–41]. $^{40}Ar/^{39}Ar$ total fusion and plateau ages for low-temperature secondary apophyllites yielded ages of 76–100 Ma from the Mesa Range in Northern Victoria Land (NVL, Fig. 1) and 95–101 Ma from Carapace Nunataks in Southern Victoria Land (SVL, Fig. 1), with a peak in the combined data at c. 95.8 Ma[40]. Rb-Sr model ages for the same apophyllites have reported ages as young as 94 Ma[40]. Furthermore, an $^{40}Ar/^{39}Ar$ plateau age of 96.7 + 0.6 Ma was determined from apophyllites for the youngest alteration event in Ferrar LIP rocks in the Prince Albert Mountains approximately ~350 km from the BRIC (Fig. 1)[39]. Hand samples and individual crystals of apophyllite were reported to have age variations of 24 and 14 Ma, respectively. Typical temperatures estimated for this reported Cretaceous alteration event range from 150 to 350 °C[17–19]. We suggest the similarity in age between BRIC vitrophyre pod samples and

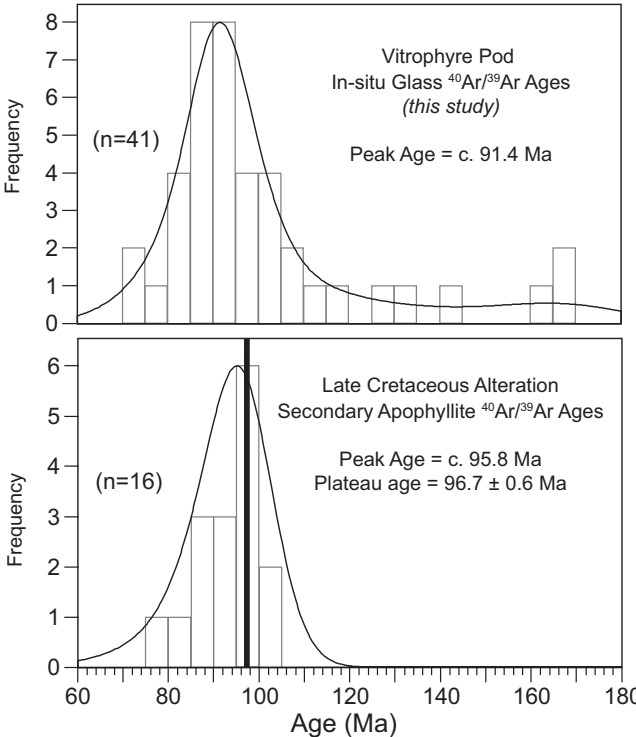

**Fig. 6 | Regional summary of Ar/Ar alteration ages from Victoria Land, Antarctica.** Kernel density plot and histogram of in situ $^{40}Ar/^{39}Ar$ ages for glass from vitrophyre pod samples of the Butcher Ridge Igneous Complex (BRIC) compared to total fusion and plateau $^{40}Ar/^{39}Ar$ ages for secondary apophyllite minerals within Jurassic Ferrar Large Igneous Province rocks from Northern and Southern Victoria Land that constrain the timing of regional Late-Cretaceous alteration[39,40].

secondary apophyllites from throughout the region is strong evidence for the reliability of vitrophyre pod ages and that the vitrophyre pod $^{40}Ar/^{39}Ar$ system had been completely reset during the Cretaceous alteration event.

These data from the BRIC are, therefore, consistent with previous data reported for the region and support a Late Cretaceous, ~72–105 Ma peak at c. 91.4 Ma, alteration event in the TAM that reset the argon system within the BRIC. Cretaceous alteration within the TAM is generally attributed to either deep burial in the Mesozoic Victoria Basin[41,42], an elevated thermal gradient combined with circulating fluids related to a failed rift, a mantle plume[39], or increased fluid circulation during uplift and denudation of the TAM[40,43]. Hydrated glass within the BRIC yields variable and spatially heterogeneous ages comparable to variations seen in apophyllites. Considering these variations, we suggest the peak age, c 91.4 Ma, is the most reliable age owing to its consistency with other Late-Cretaceous alteration ages from the region. Younger ages for vitrophyre glass and apophyllite samples, down to c. 72 and 76 Ma, may indicate that alteration was a prolonged process beginning around 100 Ma with hydration reaching near solubility limits for a majority of BRIC vitrophyre glass at c. 91.4 Ma but continuing for millions of years and ending at ~72 Ma. This hypothesis is consistent with a model for Cretaceous alteration by deep burial in the Mesozoic Victoria Basin that persisted until exhumation in the Cenozoic[41,42].

The latter observation from these data, that hydrothermal fluids involved in Cretaceous alteration of the BRIC were sourced from polar glacial melt water, relies on the long-term stability of hydrogen isotopes within hydrated glass and resistance to re-equilibration with modern or post-alteration water[22]. Consequently, the extremely depleted δD values in the BRIC glasses were either obtained during the Cretaceous or during hydrogen isotope re-equilibration that did not reset the $^{40}Ar/^{39}Ar$ age. If BRIC glasses had δD values overprinted by modern or post-Cretaceous alteration water, then we would expect similar δD values in both vitrophyre and marginal layer samples, both of which have high molecular water contents (Fig. 3). The significant difference in δD values between vitrophyre, −168 to −324 ‰, and marginal layers, −104 to −151 ‰, combined with their difference in $^{40}Ar/^{39}Ar$ age, c. 91.4 Ma versus c. 178 Ma, strongly suggests BRIC glasses avoided post-Cretaceous alteration and re-equilibration of δD by water or proton diffusion (Figs. 3, 4). Instead, their δD values coupled with field and textural observations record the composition of water during hydration. Marginal layers resisted hydration during the Cretaceous alteration event because they were already saturated with molecular water during Jurassic syn-emplacement hydration[5].

Other glassy Early Jurassic Ferrar volcanic rocks in the Mesa Range (Fig. 1), referred to as the Scarab Peak Chemical Type (SPCT), ~700 km north of the BRIC also have a Cretaceous alteration age, c. 103 Ma apophyllite Rb-Sr model age, and depleted δD values, −201 to −243 δD ‰[44]. Although these δD values are considerably higher than those in the BRIC, we argue that they are also likely indicative of polar glacial melt water. Extremely depleted δD values in rocks that are 700 km apart implies regionally extensive glaciation in Antarctica during the Cretaceous c. 90–100 Ma.

These ages coincide with the Cenomanian-Turonian thermal maximum with hot greenhouse conditions representing the warmest climate of the past 140 million years and poles conventionally thought to be ice free[9,45]. There remains considerable debate as to whether polar glaciation was possible during this time, but mounting evidence suggests polar glaciation may have occurred during the Late Cretaceous[13–19]. These data from the BRIC, therefore, provide yet another line of evidence indicating the presence of Late-Cretaceous polar glaciation in Antarctica from 72 to 105 Ma, and most likely c. 91.4 Ma during the height of hot greenhouse conditions. Furthermore, the results suggest the BRIC and associated rocks are an ideal setting in which to investigate hydration and stable isotope proxies given the significant difference between magmatic and meteoric water stable isotope compositions.

## Methods

### FTIR

For each doubly polished chip, a transmission Fourier Transform Infrared Spectroscopy (FTIR) absorbance map was generated using short-duration analyses (<1 s) on a Thermo Fisher Nicolet iN10-MX spectrometer at USGS Menlo Park. Data were collected with a 35 μm square aperture that was traced across the sample in a 50 μm grid. For each analysis, peak heights at 5230, 4520, and 3570 cm$^{-1}$ were measured. The thickness of each chip was measured with a micrometer and the wt. % water values were calculated following the methods outlined in Zhang et al.[46].

### Hydrogen isotopes

Samples of clean glass, ~1 mm in size, were separated under a microscope from any contamination products. Samples were crushed and sieved, and the 50–150 μm fraction was used for analysis as it provides the best size for water extraction[27]; Crushed material was washed several times and sonicated in deionized water and ethanol for 15 min to remove dust and particulates. Analyses were performed on a Thermo Fisher high-temperature convertor-element analyzer (TC/EA) with a glassy carbon reactor and crucibles interfaced with a Thermo MAT-253 isotope ratio mass spectrometer (IRMS) housed at the University of Oregon. Two to 3 mg of sample were loaded into Ag capsules depending on $H_2O$ content of the glass. In all cases, samples were dried overnight in a hot 130 °C vacuum drying oven. The TC/EA furnace was operated at 1450 °C and gas chromatographs were operated at 70 °C. After vacuum drying, samples were immediately placed into an autosampler carrousel under a Helium atmosphere and dropped one by one into a graphite crucible and placed in the 1450 °C furnace where they were melted to liberate $H_2O$ which was immediately quantitatively reduced with CO and $H_2$ gases. These gases pass through and are separated in gas chromatographic column held at 70 °C.

Graphite crucibles were removed between analytical sessions to reduce backgrounds and memory effects. Helium carrier gas flushes extracted volatiles to the IRMS at rates of 80–120 mL/min. Reference waters sealed in Ag tubes and micas were run concurrently for $H_2O$ and $δ^2H$ calibration. Standards run concurrently with the unknown included:

University of Oregon, Butte Montana muscovite (BUD, $δ^2H = −151‰$), as well as USGS Mica biotite (USGS57, $δ^2H$ −91‰) and muscovite (USGS58, $δ^2H$ −28‰) are used[47], and SMOW (0‰) and GISP (−195‰) waters sealed in Ag capsules. Correction for instrumental (TC/EA) mass fractionation within analytical sessions were 9–11‰ and included a linear fit between SMOW (0‰) and GISP (−195‰), as well as intermediate in $δ^2H$ values solid standards. The offsets in the lower and upper ranges were similar and so extrapolation of $δ^2H$ data to the lower $δ^2H$ range measured in this study is robust.

Water concentrations were determined by knowing water concentration (3.5 wt%) in the mica standards and used integration the peak areas of masses 2 ($H_2$) and 3 ($δ^2H$). Errors on $δ^2H$ are estimated to be in the vicinity of 2 permil while water in the vicinity of 0.05%[27,48]. Bulk water concentrations determined by TC/EA agree with bulk-rock loss on ignition (LOI) values determined by XRF.

### Oxygen isotopes

For oxygen isotope analyses we used two analytical set ups following the methods of Bindeman (2021)[30]. For triple O analyses, we fluorinated glasses and UWG2 garnet standard with a laser using $BrF_5$ reagent in a single analytical session that preceded a session where we run many mantle olivine standards.

Generated $O_2$ gas was first put through a boiling Hg to remove F2 gas excess, then it was put through an 8 ft long GC column under 10 ml/s He flow and frozen on another zeolite getter. This GC purification procedure removes NFx gases that affect mass 33 ($^{17}O/^{16}O$) during analysis as is done at the University of Oregon lab[30]. $O_2$ gas was

then run 48 times in a dual inlet mode against a well-calibrated gas to obtain absolute $D'^{17}O$ and $d^{18}O$ values on a MAT253 mass spectrometer connected to the vacuum line. Precision on $d^{18}O$ is ±0.1‰ and on $D'^{17}O$ is ±0.01‰.

$D'^{17}O$ is defined relative to a reference slope 0.5305 as:

$$D'^{17}O = d'^{17}O - 0.5305 * d'^{18}O \qquad (1)$$

Where $d'^{17}O$ are linearized as:

$$d'^{18}O = 1000 \ln(d^{18}O/1000 + 1) \qquad (2)$$

$$d'^{17}O = 1000 \ln(d^{17}O/1000 + 1) \qquad (3)$$

This procedure changes numerical values insignificantly (e.g., $d^{18}O$ is very close to $d'^{18}O$) but allows for better linear manipulation of delta values over a large delta range.

For conventional $d^{18}O$ analyses, gas generated by laser fluorination was put through boiling mercury to remove excess of $F_2$ gas, then the gas was converted to $CO_2$ by a small C-Pt converter. Yields were measured by a Baratron gauge and then $CO_2$ gas was run for one six-cycle analysis. Error on $d^{18}O$ is ±0.1‰.

## $^{40}Ar/^{39}Ar$ geochronology

All $^{40}Ar/^{39}Ar$ analytical work was performed at the University of Manitoba using a multi-collector Thermo Fisher Scientific ARGUSVI mass spectrometer, linked to a stainless-steel Thermo Fisher Scientific extraction/purification line, Photon Machines (55 W) Fusions 10.6 $CO_2$ laser and Photon Machines (Analyte Excite) 193 nm laser. Argon isotopes were measured using the following configuration: $^{40}Ar$ (H1; $1 \times 10^{12}\,\Omega$ resistor), $^{39}Ar$ (AX; $1 \times 10^{13}\,\Omega$ resistor), $^{38}Ar$ (L1; $1 \times 10^{13}\,\Omega$ resistor), $^{37}Ar$ (L2; $1 \times 10^{13}\,\Omega$ resistor) and $^{36}Ar$ (compact discrete dynode [CDD]). The sensitivity for argon measurements is ~$6.3 \times 10^{17}$ moles/fA as determined from measured aliquots of Fish Canyon Sanidine[49,50].

Standards and unknowns were placed in 2 mm deep wells in 18 mm diameter aluminum disks, with standards placed strategically so that the lateral neutron flux gradients across the disk could be evaluated. Planar regressions were fit to the standard data, and the $^{40}Ar/^{39}Ar$ neutron fluence parameter, J, interpolated for the unknowns. All specimens were irradiated in the Cadmium-lined, in-core CLICIT facility of the Oregon State University TRIGA reactor. The duration of irradiation was 35 h and using the Fish Canyon sanidine (28.201 Ma[50]).

Standards: were placed in a Cu sample tray, with a KBr cover slip, in a stainless-steel chamber with a differentially pumped ZnS viewport attached to a Thermo Fisher Scientific extraction/purification line and baked with an infrared lamp for 24 h. Single crystals were fused using the $CO_2$ laser.

Unknowns: disks, 3 mm in diameter and ~40 μm thick were cut from the same "thick" polished section investigated by optical microscopy. The disks were mounted using a ceramic adhesive (PELCO®) on a quartz slide placed in a stainless-steel chamber with a sapphire viewport attached to the same stainless-steel high vacuum extraction system as the $CO_2$ laser, and baked with an infrared lamp for 48 h. For this study, a raster size of about 100 μm × 100 μm was used and ablation pits were excavated to an estimated depth of about 40 μm.

Reactive gases were removed for both the standard and unknown, after 3 min, by three GP-50 SAES getters (two at room temperature and one at 450 °C) prior to being admitted to an ARGUS VI mass spectrometer by expansion. Five argon isotopes were measured simultaneously over a period of 6 min. Measured isotope abundances were corrected for extraction-line blanks, which were determined before every sample analysis.

Detector intercalibration (IC) between the different faraday cups was monitored (in Qtegra) every 2 days by peak hopping $^{40}Ar$. The intercalibration factor between H1 and the CDD was measured with the unknowns by online analysis of air pipettes. A value of 295.5 was used for the atmospheric $^{40}Ar/^{36}Ar$ ratio[51] for the purposes of routine measurement of mass spectrometer discrimination using air aliquots, and correction for atmospheric argon in the $^{40}Ar/^{39}Ar$ age calculation. Corrections are made for neutron-induced $^{40}Ar$ from potassium, $^{39}Ar$ and $^{36}Ar$ from calcium, and $^{36}Ar$ from chlorine[52–54]. Data collection and reduction was performed using Pychron[55]. The decay constants used were those recommended by ref. [56].

## Data availability

All data generated in this study are provided in Supplementary Data 1–4.

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

## Acknowledgements

This work was supported by National Science Foundation Graduate Research Fellowship Grant No. 1650114 (DLN) and NSF–ANT–1043152 and NSF–ANT–1443296 (J.M.C.). G. Hagen-Peter and F. Horton are

thanked for field assistance. J. Lowenstern provided access to FTIR facilities at the United States Geological Survey and insightful review prior to submission.

## Author contributions

D.L.N. and J.M.C. co-designed the project. Analyses were made by D.L.N., I.N.B., and A.C. D.L.N. and J.M.C. wrote the paper with input from I.N.B. and A.C. All authors contributed to discussion.

## Competing interests

The authors declare no competing interests.
