## [Peer Review File · Nature Communications]

Ultra-depleted hydrogen isotopes in hydrated glass record
Late Cretaceous Antarctic glaciationREVIEWER COMMENTS

Reviewer #1 (Remarks to the Author):

This manuscript presents an interesting dataset of hydrogen isotopes, oxygen isotopes, and $^{40}\text{Ar}/^{39}\text{Ar}$ ages from hydrated silicic glass from the Early Jurassic Butcher Ridge Igneous Complex (Antarctica). I thought the manuscript was well written and easy to follow. In general, the data presented seem robust and I found the arguments that the hydrogen and oxygen isotope data from the hydrated glass preserves paleoclimate information to be convincing. I found the argument that the hydration of the volcanic glass involved polar glacial melt water to be reasonable. I also found the final section of the paper to do a good job of placing these data into a broader context with implications for possible extensive glaciation in Antarctica during the Cretaceous, which contrasts with other recent studies. Overall, this is very interesting and exciting study.

The main problem I see with the manuscript is the $^{40}\text{Ar}/^{39}\text{Ar}$ dating. For the $^{40}\text{Ar}/^{39}\text{Ar}$ data presented in the manuscript to have significance, the reader must accept that the $^{40}\text{Ar}/^{39}\text{Ar}$ ages reflect the age of secondary hydration and alteration. Specifically, the ages must reflect the timing of the same glass hydration/alteration event recorded by hydrogen and oxygen isotopes. I have several questions and comments regarding this, described below:

1. The authors mention that secondary hydration and thermal alteration may promote argon loss and reset the $^{40}\text{Ar}/^{39}\text{Ar}$ age of the rock to the timing of alteration and cite Cerlin et al. (1985) to support this claim. However, Cerlin et al. (1985) does not really support this claim from what I can tell. The last sentence of the abstract for this paper is "In these glasses, Ar appears to be less mobile than alkali ions, resulting in discrepant K-Ar ages for glass with significant hydration and oxygen isotope exchange." This seems to argue against the use of hydrated glass to constrain the age of eruption or alteration because Ar and K may be affected differently by the alteration process. I don't see that Cerlin et al. (1985) claims that the $^{40}\text{Ar}/^{39}\text{Ar}$ system is totally reset (i.e., all radiogenic ^{40}Ar removed from the system), which is essential if these ages are to accurately date alteration events. Lastly, the authors suggest that if younger alteration had happened then we would expect to see younger $^{40}\text{Ar}/^{39}\text{Ar}$ ages. However, the paper the authors cite shows that in many cases the most altered samples have older ages due to preferential K-loss (relative to Ar). Because of this I don't feel the statement that younger alteration would result in younger ages (or a larger range of ages than is observed) is well defended as currently written. I think the authors need to do more to back up the claim that these $^{40}\text{Ar}/^{39}\text{Ar}$ ages can reasonably be interpreted to reflect the timing of alteration instead of simply disturbed ages due to Ar and K mobility.

2. Related to the first point, what temperature is this alteration thought to have happened at? A temperature range of 150 oC to 350oC is mentioned in the final section of the paper for other places

within the TAM, but I wasn't sure if that is also thought to be the case for the BRIC. Essentially, I'm curious because it seems important to the assumption that the $^{40}\text{Ar}/^{39}\text{Ar}$ system had been reset.

3. The way the $^{40}\text{Ar}/^{39}\text{Ar}$ data are handled is another potential problem, or at least something that should be addressed. Most samples display a wide range of ages and samples yield different averages or modes of ages when considered individually. However, in the final interpretation of the $^{40}\text{Ar}/^{39}\text{Ar}$ ages, data from all the samples are considered together to define the peak at 91.4 Ma. I think this is an oversimplification of the data. The $^{40}\text{Ar}/^{39}\text{Ar}$ ages are not well represented by a single age given that the dispersion of ages is far more than can be explained by analytical uncertainty. This to me suggests that either the ages are inaccurate or that hydrothermal alteration extended down to 70 Ma or so. It may be fair to say that the alteration in the glasses you analyzed is broadly consistent with this 90 - 130 Ma alteration event (assuming the ages do reflect alteration ages), but I think it is inaccurate to only attribute it to one event centered around 91.4 Ma. Perhaps the $^{40}\text{Ar}/^{39}\text{Ar}$ ages are better described as a range of ages.

4. Lastly, the dispersion in the $^{40}\text{Ar}/^{39}\text{Ar}$ ages measured for the vitrophyre samples could be taken as evidence that the $^{40}\text{Ar}/^{39}\text{Ar}$ system was not completely reset during hydration/alteration. If this is the case, then how can one be sure that the 91.4 Ma age has significance?

In summary, I think if a stronger case can be made for why the $^{40}\text{Ar}/^{39}\text{Ar}$ ages reliably record the age of secondary hydration, or if the authors can recast the discussion to better highlight how the story is still interesting even if the $^{40}\text{Ar}/^{39}\text{Ar}$ ages are not accurate, then this manuscript would be in good shape for publication. In addition to these comments, I have included some minor suggestions in the annotated PDF.

Best of luck with the revisions,

Mark Stelten

Reviewer #2 (Remarks to the Author):

The manuscript "Ultra-depleted hydrogen isotopes in hydrated glass record Cretaceous glaciation in Antarctica" by Nelson et al. displays new D/H values for aqueously altered rhyolitic glasses from the early Jurassic BRIC in the transantarctic Mountains. These data, coupled to Ar/Ar ages, are then used to profoundly revise the temperature range in Antarctica during the Jurassic. This is a very well-written and convincing manuscript; the line of reasoning is straight and simple; the conclusion sounds. I have only minor comments that can be summarized in one line: The Sup. Online do not provide the necessary analytical details and the main text should provide more detailed information in several places. As it is, the paper does not read as a stand-alone paper and non-specialists will have constantly refer to literature from the authors to find short yet important information. This should not be the case.

Beside these relatively minor (or moderate) improvements, this manuscript could be published by Nat. Com. and will have a strong impact on a large community of Earth Scientists.

From the start, it would be interesting to present the published literature and arguments calling for a warm Antarctica during Cretaceous. The major conclusion of this manuscript is that, it was very cold (water very depleted in D). But it is actually difficult to find in the text references claiming the opposite and the basis for that. Please provide more background on this aspect.

Line 62 and after: There is always both OH and H₂O in hydrous silicate glasses and melts, especially at saturation. So, the statement is oversimplistic. This speciation (in glasses and melts) depends on a sum of parameters that do not need to be reviewed, but it seems important to me to refer to some pivotal literature and to provide a more balanced description of this aspect. Along the same line, the difference in mobility of these two species should be mentioned somewhere as the lability of water molecules is important in the present context.

Line 71: More details here and an analytical Sup. Online providing the methods, the nature of the apparatuses, the spatial and spectral resolution/calibration etc... All the basic information concerning the data collected in this study must be found in this manuscript and in the associated Sup. Online.

Lines 73 and after: Please discuss this in the light of the literature concerning the solubility limit in these magmas and make this section more robust from this point of view.

Line 95: Would it be possible to provide basic mineralogical information here instead of using the generic word "Minerals"? Again, these lines sound as broad general statements where I am certain that a short discussion of the mineralogy, of the water solubility and speciation in dominant phases would be much stronger.

Line 105: Please define D'17O. As for the FTIR, analytical methods are mandatory for the D/H analysis. How, where, on what sample alicot were these measurements performed how was water extracted, is there a dependence of the D/H as a function of the OH/H₂O measured by FTIR?... A proper Sup Online and a few sentences in the main text is required.

Line 113: This is true but it still depends on the speciation of water and the phases that accommodate water. I certainly agree with the author, but as it is written it sounds like there is never significant kinetic D/H fractionation during kinetically-driven processes. This is not exact. In nominally anhydrous minerals (NAM), when protons are the dominant diffusing species (or molecular H₂), the fractionation can be as large as several hundreds of permil. Here the conclusion seems robust because the redox conditions do

not seem to be reducing enough to allow H or H₂ to be a dominant species. The main carriers of water (not perfectly identified or discussed) are certainly not NAMs. Still, I would like, if possible, to have a more accurate description of this aspect.

Line 155: Here, there is a proper method section. I would just like to find some information on the phases that accommodate the Ar and what are the closure temperatures and the behavior of these phases during aqueous alteration. Would Ar/Ar data provide an age for the last pervasive aqueous alteration or the last moderate circulation of water or the age of the alteration of the mineral carriers (and not necessarily the whole rock). Do the Ar-bearing phases survive to several episodes of aqueous alteration? Please provide a petro-geochemical interpretation of these data and assess the fact that ages given by Ar/Ar are truly those of the bulk outcrop aqueous alteration.

Reviewer #3 (Remarks to the Author):

This paper aims at reconstructing the paleoclimate of Antarctica during the Late Cretaceous using ancient meteoric water preserved in volcanic glass shards. The water concentrations have been measured on different rhyolite samples (vitrophyre pod, vitrophyre layers and marginal layers) using a Fourier transform infrared spectroscope. The signature of the fluids present during hydration of the volcanic glasses have been determined by measuring the hydrogen isotope ratios and triple oxygen isotopes of the same samples. In addition, ⁴⁰Ar/³⁹Ar geochronology has been used to determine the timing of volcanic glass hydration. Results indicate that volcanic glasses have interacted with depleted polar glacial melt water at ca. 90 Ma which is in contradiction with warm weather conditions suggested by previous authors.

Although this study presents a good preliminary dataset providing new information on the paleoclimate of Antarctica, it contains some weaknesses. First, the description of the studied samples at the microscopic scale is poor and not supported by thin section photographs (Lines 46-51). Second, the calculations of meteoric water reconstructions are not explained in detail making things very confusing (e.g. equations, temperature of fractionation between glasses and water,...)...Third, references are missing (e.g. Line 45, Line 94, Line 116, Line 169) making the text weak. Fourth, the ¹⁸O/¹⁶O and ¹⁷O/¹⁶O ratios are not shown/clearly written in the supplementary table 2 as well as the standard deviation for each ratio. Fifth, the text is not always very well-structured and lacks precise vocabulary making things difficult to understand. We have the impression that part of the information is missing and that the authors expect the readers to master the methods used for this study. In addition, localities mentioned in the text must appear on Figures (Mc Murdo Station).

Therefore, based also on the fifth main points mentioned above, I am afraid that this manuscript cannot be accepted for publication in Nature Communications as it stands. I am happy to look at a new version of the manuscript.

Kind Regards,

Aude Gébelin

Our replies to reviewers are given below in blue.

Reviewer #1 (Remarks to the Author):

1. The authors mention that secondary hydration and thermal alteration may promote argon loss and reset the $^{40}\text{Ar}/^{39}\text{Ar}$ age of the rock to the timing of alteration and cite Cerlin et al. (1985) to support this claim. However, Cerlin et al. (1985) does not really support this claim from what I can tell. The last sentence of the abstract for this paper is "In these glasses, Ar appears to be less mobile than alkali ions, resulting in discrepant K-Ar ages for glass with significant hydration and oxygen isotope exchange." This seems to argue against the use of hydrated glass to constrain the age of eruption or alteration because Ar and K may be affected differently by the alteration process. I don't see that Cerlin et al. (1985) claims that the $^{40}\text{Ar}/^{39}\text{Ar}$ system is totally reset (i.e., all radiogenic ^{40}Ar removed from the system), which is essential if these ages are to accurately date alteration events.

Lastly, the authors suggest that if younger alteration had happened then we would expect to see younger $^{40}\text{Ar}/^{39}\text{Ar}$ ages. However, the paper the authors cite shows that in many cases the most altered samples have older ages due to preferential K-loss (relative to Ar). Because of this I don't feel the statement that younger alteration would result in younger ages (or a larger range of ages than is observed) is well defended as currently written. I think the authors need to do more to back up the claim that these $^{40}\text{Ar}/^{39}\text{Ar}$ ages can reasonably be interpreted to reflect the timing of alteration instead of simply disturbed ages due to Ar and K mobility.

We only intended to cite Cerlin as foundational research on this topic and not to indicate that hydration/alteration always favors argon loss/resetting. We have revised the text to clarify the findings of Cerlin and that argon loss or potassium loss can occur. We have added additional references as foundational research on this topic. We also note that because the ages are essentially uniformly younger than emplacement, the alteration process appears to have favored argon loss, increasing the potential for a partial or complete age reset. We have added additional references and text to more clearly explain this. We have also added additional text to reinforce that the primary line of evidence for the reliability of these alteration ages is the consistency with regional alteration ages from multiple localities utilizing multiple methods.

2. Related to the first point, what temperature is this alteration thought to have happened at? A temperature range of 150 oC to 350oC is mentioned in the final section of the paper for other places within the TAM, but I wasn't sure if that is also thought to be the case for the BRIC. Essentially, I'm curious because it seems important to the assumption that the $^{40}\text{Ar}/^{39}\text{Ar}$ system had been reset.

We have added text to indicate the assumed temperatures of hydration are likely to be 175 to 375°C based on comparisons with solubility experiments and consistent with the regional alteration temperatures referenced, 150-350 degrees C. It is our assumption that hydration occurred below the Ar closure temperature within glass and minerals, i.e., $< \sim 500$ oC. Therefore, the primary mechanism for resetting of the Ar system is hydration (which is facilitated by elevated but still low to moderate temperatures).

3. The way the $^{40}\text{Ar}/^{39}\text{Ar}$ data are handled is another potential problem, or at least something that

should be addressed. Most samples display a wide range of ages and samples yield different averages or modes of ages when considered individually. However, in the final interpretation of the $^{40}\text{Ar}/^{39}\text{Ar}$ ages, data from all the samples are considered together to define the peak at 91.4 Ma. I think this is an oversimplification of the data. The $^{40}\text{Ar}/^{39}\text{Ar}$ ages are not well represented by a single age given that the dispersion of ages is far more than can be explained by analytical uncertainty. This to me suggests that either the ages are inaccurate or that hydrothermal alteration extended down to 70 Ma or so. It may be fair to say that the alteration in the glasses you analyzed is broadly consistent with this 90 - 130 Ma alteration event (assuming the ages do reflect alteration ages), but I think it is inaccurate to only attribute it to one event centered around 91.4 Ma. Perhaps the $^{40}\text{Ar}/^{39}\text{Ar}$ ages are better described as a range of ages.

We have revised the manuscript to emphasize the age range from 72 to 105 Ma with a peak at c. 91.4 Ma as per the reviewer's comment. We have also included additional discussion and a figure to compare these new Ar data to regional alteration Ar data to help demonstrate the consistency between the data. Our intention is not to attribute it to one event centered at 91.4 Ma but we emphasize the age considering it represents a significant clustering of the data, it is the peak of the distribution, and it is consistent with other ~90-100 Ma published alteration ages. We now include text to explain that we interpret the ages to represent a protracted hydration process where hydration of the glass reached saturation in the majority of the BRIC vitrophyre around c. 91.4 Ma but may have continued to c. 72 Ma. This hypothesis is consistent with current models for the Cretaceous alteration event and with apophyllite ages that cluster around 96 Ma but range down to c. 76 Ma.

4. Lastly, the dispersion in the $^{40}\text{Ar}/^{39}\text{Ar}$ ages measured for the vitrophyre samples could be taken as evidence that the $^{40}\text{Ar}/^{39}\text{Ar}$ system was not completely reset during hydration/alteration. If this is the case, then how can one be sure that the 91.4 Ma age has significance?

We rely heavily on the several studies reporting Ar/Ar data for secondary apophyllite and whole-rock samples from other Jurassic Ferrar LIP volcanic rocks from several localities that demonstrate nearly indistinguishable ages for alteration to those reported in this study. In the absence of these data, it may be challenging to confidently state the significance of the age range from 72 to 105 Ma and the peak at 91.4 Ma. However, we suggest the consistency between published dates and our new data is the strongest evidence for the significance and reliability of our reported ages, especially considering that they are mutually exclusive methods, i.e., Ar-Ar on glass versus Ar-Ar on apophyllite versus Rb-Sr on apophyllite all largely in agreement (including the dispersion/variation). We have revised the manuscript to better make this case.

In summary, I think if a stronger case can be made for why the $^{40}\text{Ar}/^{39}\text{Ar}$ ages reliably record the age of secondary hydration, or if the authors can recast the discussion to better highlight how the story is still interesting even if the $^{40}\text{Ar}/^{39}\text{Ar}$ ages are not accurate, then this manuscript would be in good shape for publication. In addition to these comments, I have included some minor suggestions in the annotated PDF.

Reviewer 1 comments from PDF:

Page 3 Text in panel A is hard to read, i.e.. "Transantarctic Mountains"

Font size increased

Page 3 I'd show, or mention, where in Figure 1 these photos are from.

Locations of (C) and (D) now on Fig. 1B.

Page 7 Is this plot shown somewhere? If not, I'd include it since you mention it.

We have chosen not to add the plot since it distracts from the main message of the manuscript.

Page 7 It would be good to have a ballpark estimate of how significant this effect would be. What do you think the "minor" contribution would be?

Additional text added

Page 9 Why do you think this vitrophyre does not have ages down to 91.4 Ma like the other vitrophyre?

Additional text added

Page 9 delete vitrophyre, replace with vitrophyre pod

Changed as suggested

Page 10 I'm confused by this sentence. You are discussing the isotope compositions then switch to water contents at the end. How do the water contents relate to your point about the difference in isotope composition?

Additional text added

Page 11 Delete preliminary

Deleted as suggested

Mark Stelten

Reviewer #2 (Remarks to the Author):

The Sup. Online do not provide the necessary analytical details and the main text should provide more detailed information in several places. As it is, the paper does not read as a stand-alone paper and non-specialists will have constantly refer to literature from the authors to find short yet important information. This should not be the case.

From the start, it would be interesting to present the published literature and arguments calling for a warm Antarctica during Cretaceous. The major conclusion of this manuscript is that, it was very cold (water very depleted in D). But it is actually difficult to find in the text references claiming the opposite and the basis for that. Please provide more background on this aspect.

We have revised the manuscript to include summary statements on this topic with several references within the introduction to highlight and emphasize this discrepancy. We have also revised the text near the end of the manuscript. In both cases we have added more references to better contextualize this topic/debate within the paleoclimate research community and underscore that the hypothesis for ice-free poles during the Late Cretaceous has been conventional thought for decades:

“It is a well-established fact that no polar ice existed during the Paleocene-Eocene Thermal Maximum or the Cenomanian-Turonian Thermal Maximum (93 Ma; Zielgler et al., 1985).”

Scotese, CR, Song, H, Mills, BJW orcid.org/0000-0002-9141-0931 et al. (2021) Phanerozoic Paleotemperatures: The Earth’s Changing Climate during the Last 540 million years. *Earth-Science Reviews*, 215. 103503. ISSN 0012-8252

<https://doi.org/10.1016/j.earscirev.2021.103503>

Line 62 and after: There is always both OH and H₂O in hydrous silicate glasses and melts, especially at saturation. So, the statement is oversimplistic. This speciation (in glasses and melts) depends on a sum of parameters that do not need to be reviewed, but it seems important to me to refer to some pivotal literature and to provide a more balanced description of this aspect. Along the same line, the difference in mobility of these two species should be mentioned somewhere as the lability of water molecules is important in the present context.

We have added text to note the complexity of water speciation in hydrous glasses. Several references for foundational research on the topic have also been added in support of the statements made.

Line 71: More details here and an analytical Sup. Online providing the methods, the nature of the apparatuses, the spatial and spectral resolution/calibration etc... All the basic information concerning the data collected in this study must be found in this manuscript and in the associated Sup. Online.

FTIR analytical methods have been added to supplementary files.

Lines 73 and after: Please discuss this in the light of the literature concerning the solubility limit in these magmas and make this section more robust from this point of view.

Added Hudak reference for solubility of rhyolitic glass at temperatures expected in this case.

Line 95: Would it be possible to provide basic mineralogical information here instead of using the generic word “Minerals”? Again, these lines sound as broad general statements where I am certain that a short discussion of the mineralogy, of the water solubility and speciation in dominant phases would be much stronger.

Added a brief description of the mineralogy to the introduction that describes the dominant phases/mineralogy.

Line 105: Please define D'17O. As for the FTIR, analytical methods are mandatory for the D/H analysis. How, where, on what sample alicot were these measurements performed how was water extracted, is there a dependence of the D/H as a function of the OH/H₂O measured by FTIR?... A proper Sup Online and a few sentences in the main text is required.

FTIR and DH analytical methods supplementary text has been added.

Line 113: This is true but it still depends on the speciation of water and the phases that accommodate water. I certainly agree with the author, but as it is written it sounds like there is never significant kinetic D/H fractionation during kinetically-driven processes. This is not exact. In nominally anhydrous minerals (NAM), when protons are the dominant diffusing species (or molecular H₂), the fractionation can be as large as several hundreds of permil. Here the conclusion seems robust because the redox conditions do not seem to be reducing enough to allow H or H₂ to be a dominant species. The main carriers of water (not perfectly identified or discussed) are certainly not NAMs. Still, I would like, if possible, to have a more accurate description of this aspect.

We have tried to be clear throughout the manuscript that the main carrier of water within the BRIC is the volcanic glass. To do so we have provided FTIR for glass and bulk rock H₂O data that, when compared, indicate glass as the carrier. We have revised the manuscript to emphasize the mineralogy of the samples to clarify this point.

The reviewer's comment does not indicate a disagreement but rather requests that we discuss kinetic fractionation within NAMs in addition to glass. If NAMs were anticipated to impact these measurements, we would welcome the discussion but in this case the demonstrated carrier of water is the glass, so much so that the hydrogen isotope composition is overwhelmingly controlled by the glass. We consider a discussion of kinetic fractionation within NAMs to unnecessarily complicate this issue for readers.

Line 155: Here, there is a proper method section. I would just like to find some information on the phases that accommodate the Ar and what are the closure temperatures and the behavior of these phases during aqueous alteration. Would Ar/Ar data provide an age for the last pervasive aqueous alteration or the last moderate circulation of water or the age of the alteration of the mineral carriers (and not necessarily the whole rock). Do the Ar-bearing phases survive to several episodes of aqueous alteration? Please provide a petro-geochemical interpretation of these data and assess the fact that ages given by Ar/Ar are truly those of the bulk outcrop aqueous alteration.

We have revised the text of the Ar section to clarify that we are concerned almost exclusively with the Ar/Ar age of the glass, the minerals (which we have added a description of, plag+px+ox+kspar+qtz) are all expected to be ~183 Ma, the known emplacement age of the BRIC. We have also included reference to the estimated closure temperature ~500 °C and that hydration/alteration is expected to have occurred at a lower temperature and therefore the mechanism for disturbing the Ar system in glass is through alteration/substitution of the glass structure during hydration. If the Ar system within the glass had been completely reset during secondary hydration (at or near the solubility of water in glass) the data would represent the last time BRIC had undergone pervasive alteration/hydration. The minerals appear to have survived the low-temperature alteration but the glass should become hydrated to the

solubility limit and no further hydration should occur. Devitrification of the glass would be expected eventually and it remains unclear why this hydration event did not promote more pervasive devitrification. Our revisions to the text clarify that the Ar ages reflect bulk outcrop hydration of BRIC glasses, as is the intention of our manuscript.

We have added additional references towards these ends.

Reviewer #3 (Remarks to the Author):

First, the description of the studied samples at the microscopic scale is poor and not supported by thin section photographs (Lines 46-51).

We have added a brief description of the mineralogy of the samples to the introduction. Detailed petrographic descriptions and photomicrographs as well as SEM images for the studied samples are provided in Nelson et al. 2020 and we prefer to not reproduce the information here.

Second, the calculations of meteoric water reconstructions are not explained in detail making things very confusing (e.g. equations, temperature of fractionation between glasses and water,...)...

We have added an FTIR methods section that describes how water was determined.

Third, references are missing (e.g. Line 45, Line 94, Line 116, Line 169) making the text weak.

References were added for lines 45, 94, and 116. Line 169 and the subsequent sentence are the same references, we included the references to both sentences.

Fourth, the $^{18}\text{O}/^{16}\text{O}$ and $^{17}\text{O}/^{16}\text{O}$ ratios are not shown/clearly written in the supplementary table 2 as well as the standard deviation for each ratio.

Revised supplementary table accordingly.

Fifth, the text is not always very well-structured and lacks precise vocabulary making things difficult to understand. We have the impression that part of the information is missing and that the authors expect the readers to master the methods used for this study.

We have revised the entire manuscript to address these issues.

In addition, localities mentioned in the text must appear on Figures (Mc Murdo Station).

We have added a new overview map that includes all of the localities mentioned in the text.

Aude Gébelin

REVIEWERS' COMMENTS

Reviewer #1 (Remarks to the Author):

After reading the responses to reviewers comments and re-reading the manuscript, I feel that the comments I provided have been adequately addressed. I think the expanded discussion (and Fig. 6) help provide context and strengthen the argument for the interpretation of the $^{40}\text{Ar}/^{39}\text{Ar}$ data.

I have added some suggestions to the discussion for areas where the text could be clarified (in the attached PDF).

I think the manuscript is suitable for publication after some rewriting of the discussion to improve clarity.

Reviewer #2 (Remarks to the Author):

The authors have addressed most of previous reviewers concerns with the original manuscript, including mine. I believe the quality of this manuscript has been significantly improved. The context and the useful analytical details have been added. Some interpretations and conclusions have been tempered (yet they are convincing and of broad significance) providing a more balanced view of the controversy, open up the door to additional analysis of competing groups in the future. Given the large implications of this study, I recommend publication of this manuscript in Nat Com.

Mathieu Roskosz

Reviewer #3 (Remarks to the Author):

Dear editor, dear authors,

the new version of the manuscript has been improved and includes additional relevant references, analytical methods, sample description, and temperature that prevailed during volcanic glass-water interactions. I also like the discussion on the timing of rock alteration (was needed indeed)!

I may not have been very precise in my review comments but although the supplementary analytical methods contain information on FTIR, hydrogen, and oxygen isotopes and additional info on the triple oxygen isotope .xls table, it would have been nice to include a diagram showing the $\delta^{18}\text{O}$ - $\delta^{17}\text{O}$ relationship...Anyways...

I am happy to accept this manuscript for publication in Nature communications. That's good science!

All the best,

Aude

Reviewer #1 responses:

Line 188: Meaning the same as the hydrogen isotope/bulk water samples, or that all the Ar samples were from one outcrop?

We changed to clarify that we are referring to the same samples as the hydrogen isotope/bulk water samples

Line 191: Does this refer to the vitrophyre pod, vitrophyre layer, or both? I would make this more clear. Agreed that this might be confusing. We have re-worded to clarify.

Line 193: add “,”

Added as suggested

Line 194: delete “,”

Deleted as suggested

Line 194: I'm confused as to what you are referring to here. I thought the date range for the vitrophyre layer was 190 Ma to 137 Ma. Also, didn't all the data for the vitrophyre layer fall within the range?

This has been re-worded to clarify that we are referring to dates between the emplacement age and the main peak of alteration at ~90 Ma.

This seems like a reasonable interpretation. out of curiosity, is there any evidence from the K/Ca of the analysis that would indicate minerals were included?

We did look at these data, but see no trends in the K/Ca or K/Cl ratios to indicate inclusion of minerals.

Line 197: Perhaps highlight that this is close to the emplacement age, suggesting minimal resetting.

Changed as suggested

Line 221: Needs citation.

Citation added

Line 226: I think this could be cut as it was highlighted elsewhere.

We would prefer to leave this in as it is not directly stated in relation to the regional alteration data